# Risk Factors for the Anal and Oral Human Papillomavirus (HPV) Infections among Women with Severe Cervical Lesions: A Prospective Case—Control Study

**DOI:** 10.3390/biomedicines11123183

**Published:** 2023-11-29

**Authors:** Monika Nipčová Džundová, Borek Sehnal, Michal Zikán, Roman Kocián, Olga Dubová, Petr Hubka, Lukáš Dostálek, Pavel Kabele, Tomáš Brtnický, Jiri Slama

**Affiliations:** 1Department of Gynecology and Obstetrics, Bulovka University Hospital, 1st Faculty of Medicine, Charles University, 18081 Prague, Czech Republic; monika.nipcova@bulovka.cz (M.N.D.); michal.zikan@lf1.cuni.cz (M.Z.); olga.dubova@bulovka.cz (O.D.); petr.hubka@bulovka.cz (P.H.); pavel.kabele@bulovka.cz (P.K.); tomas.brtnicky@bulovka.cz (T.B.); 2Department of Gynecology, Obstetrics and Neonatology, General University Hospital, 1st Faculty of Medicine, Charles University, 12808 Prague, Czech Republic; roman.kocian@vfn.cz (R.K.); lukas.dostalek@vfn.cz (L.D.); jiri.slama@vfn.cz (J.S.)

**Keywords:** human papillomavirus, HPV infection, anal HPV, oral HPV, risk factors, CIN2+ patients

## Abstract

The carcinogenicity of HPV infection in the anogenital and oropharyngeal regions is broadly accepted. The aim of the study was to define risk factors for anal and oral HPV infections in high-risk patients with biopsy-proven severe cervical lesions (CIN2+). Altogether immunocompetent 473 females with CIN2+ were categorized into the study group and another 245 women into the control group. The strongest risk factor for anal HPV infection was the presence of cervical HPV infection (*p* < 0.001). Furthermore, ten or more lifetime sexual partners (*p* = 0.013), a sexual non-coital contact with the anal area (*p* < 0.001), and actively practicing anal-penetrative intercourse (*p* < 0.001) were significantly associated with anal HPV. A history of genital warts in the woman (*p* = 0.010) and the presence of genital warts in the male partner (*p* = 0.029) were found statistically significant for the risk of oral HPV infection. Our data suggest that the presence of HPV infection, especially high-risk genotypes, in one anatomical site poses the greatest risk for HPV infection in another anatomical site. The cervix is the main reservoir of infection, but the risk factors for anal and oral HPV infections are dissimilar according to different anatomical distances and more complex routes of transmission.

## 1. Introduction

Human papillomavirus (HPV) infection is broadly accepted to be the most common sexually transmitted infection worldwide [1]. HPV can cause a substantial proportion of benign, premalignant, and malignant tumors in the anogenital and oropharyngeal regions [1]. The total number of HPV-related cancers has been globally estimated to include approximately 4.5% of all human cancers every year but 8.6% in women and 0.8% in men [2,3]. In 2018, the age-standardized global incidence of HPV-related cancers was estimated at 8.0 per 100,000 persons [2]. An estimated 690,000 new HPV-associated cancers occurred worldwide in 2018 [2]. Cervical cancer accounts for 83% of all HPV-attributable cancers, two-thirds of which occur in less-developed countries [2,3]. Cervical cancer is the fourth most frequently diagnosed cancer in women, with an estimated 604,000 new cases and 342,000 deaths worldwide in 2020 [4]. HPV infection is responsible for almost all cervical cancers and for approximately 88% of anal and 38% of oropharyngeal cancers [3]. While cervical cancers occur mainly in developing countries, the increasing prevalence of HPV-related head and neck cancers is a greater problem in developed countries. The global prevalence of cervical HPV infection in the general population of women with normal cervical cytology has been estimated to be 10–12%, with a significantly higher rate in Eastern Europe (21%) [1,5]. The prevalence of anal HPV infection in women is higher than in heterosexual men. Large studies have found a prevalence of anal HPV infection in 27–31% of healthy sexually active women [6,7,8,9]. On the other hand, the overall prevalence of oral HPV infections in the healthy European population ranges widely from 1.2–11.6% [10]. Every anatomical site has different risk factors for the prevalence of HPV infections. Various practices of risky sexual behavior, unhealthy lifestyle, and alcohol and tobacco use are most often mentioned [1,5,6,7,8,9]. The knowledge about risk factors for HPV infections at different anatomical localizations is essential for the definition of particular risks for developing HPV-associated malign and benign lesions. The aim of our study was to define risk factors for the presence of anal and oral HPV infections in high-risk patients with biopsy-proven severe cervical lesions. Such a study has not yet been performed until now.

## 2. Materials and Methods

### 2.1. Patients

A total of 718 females were recruited from patients attending one of the participating university-based colposcopy clinics collaborating with the First Medical School of Charles University in Prague. Women were familiarized with the study protocol, and they were excluded if they were HIV-positive, pregnant, or unable to give informed consent. First, an anonymous self-administered questionnaire including questions about age, education, social status, marital status, sexual behavior (including unprotected vaginal, non-coital contact to anus, anal intercourse, and oral sex), reproductive history, use of hormones, smoking habits, medical history, and family history was administered. The cervical, anal, and oral samples for the HPV-genotyping test of the same patient were obtained at the same time. The study group consisted of high-risk women with histologically proven high-grade squamous or glandular intraepithelial lesion or microinvasive cervical cancer (cervical intraepithelial neoplasia grade two or worse, CIN2+). The control group represented low-risk women without CIN2+. The study was approved by the Local Ethical Committee under judgement’s reference number 1862012/6233/EK-Z.

### 2.2. HPV Detection and Genotyping

Cytobrush smears from the cervix or from the anus and oral rinses with sterile distilled water for 30 s were used to collect each sample. Each cytobrush was immersed, and each oral rinse was placed in a different vial containing Cobas PCR Cell Collection medium to allow cell transport and preservation of cells. The Linear Array HPV Genotyping Test (Roche Molecular Systems, Inc., Branchburg, NJ, USA) was subsequently used according to the producer’s instructions to identify DNA from 37 HPV genotypes that included 13 high-risk (HPV 16, 18, 31, 33, 35, 39, 45, 51, 52, 56, 58, 59, and 68) and 24 low-risk types (HPV 6, 11, 26, 40, 42, 53, 54, 55, 61, 62, 64, 66, 67, 69, 70, 71, 72, 73, 81, 82, 83, 84, IS39, and CP6108).

### 2.3. Histopathology

All biopsy specimens submitted for histological assessment were routinely examined in their entirety. Sections from the formalin-fixed and paraffin-embedded tissue fragments were stained with hematoxylin–eosin. Histological grading of high-grade dysplasia was based on the standard criteria.

### 2.4. Statistics

The standard robust summary statistics were applied to describe primary data, absolute and relative frequencies for categorical variables, and median as supplied with the 5th–95th percentile range for continuous variables. The statistical significance of differences between the group of subjects with anal or oral HPV infections and the group of subjects without anal or oral HPV infections was tested in categorical variables using Fisher’s exact test; an exact Monte Carlo method with 100,000 samples was applied to estimate the significance of differences in variables with more than two categories. The Mann–Whitney U-test was applied to test the differences in continuous variables. Age-adjusted logistic regression analysis was used for the assessment of the association between various risk factors and defined end-points related to different types of HPV infection. The results are represented as estimates of odd ratios (OR, along with 95% confidence interval) with corresponding statistical significance (Wald’s test). All analyses were performed using SPSS 24.0.1 (IBM Corporation, New York, NY, USA, 2016).

## 3. Results

The entire cohort included a total of 718 immunocompetent women. Altogether high-risk 473 females (175 with confirmed CIN 2, 254 with CIN 3, and 44 with microinvasive cervical cancer) were categorized into the study group and low-risk 245 women into the control group. Table 1 shows the demographic characteristics of the whole studied population with some statistically significant differences between both groups.

HPV infection was detected in 62.7% (448/714) of cervical, 36.6% (246/673) of anal, and 2.3% (10/438) of oral samples sufficient for HPV testing. The prevalence of cervical (81.4% vs. 26.9%; *p* < 0.001) and anal (43.3% vs. 24.5%; *p* < 0.001) HPV infection was significantly higher in the study group than in the control group, but the difference in the prevalence of oral (2.7% vs. 1.4%; *p* = 0.511) HPV infection was not statistically significant between the two groups. HPV infection was not detected in any anatomical site in 9.5% (45/473) of subjects in the study group and in 34.3% (84/245) of the controls (*p* < 0.001).

The prevalence of HPV infections at different anatomical sites, including concurrent infections, with *p*-value between groups is shown in [Fig biomedicines-11-03183-ch001].

The observed risk factors for the prevalence of anal and oral HPV infections were dissimilar for different anatomical sites. The strongest risk factor for the prevalence of anal HPV infection was the presence of cervical HPV infection (*p* < 0.001).

There were determined to be three strong risk factors for the prevalence of anal HPV infection: ten or more lifetime sexual partners (*p* = 0.013), report of a sexual non-coital contact with the anal area (*p* < 0.001), and active practice of anal-penetrative intercourse (*p* < 0.001). Women reporting hormone use (*p* = 0.028) or smoking (*p* = 0.046) were significantly associated with anal HPV infection, too. On the other hand, a history of more than two pregnancies (*p* = 0.045) and more than one delivery (*p* = 0.013) and age > 50 years (*p* = 0.048) were statistically significant protective factors for the prevalence of anal HPV infection (Table 2).

We defined three statistically significant risk factors for oral HPV infection, but there were only eight oral HPV infections among high-risk women. However, HR genotypes were detected in all HPV-positive oral samples of patients with CIN2+. A history of genital warts in the woman (*p* = 0.010) and also the presence of genital warts in the male partner (*p* = 0.029) were found statistically significant for the prevalence of oral HPV infection. Surprisingly, at least one pregnancy (*p* = 0.005) but no history of a delivery was detected as a significant risk factor for oral HPV (Table 3).

## 4. Discussion

As expected, we found a higher prevalence of HPV infection at all examined anatomical sites in the study group among high-risk women with CIN2+ compared to the women in the control group. The difference was statistically significant in the anus (*p* < 0.001) but not in the oral cavity (*p* = 0.511) according to small whole number of oral-HPV-positive cases. The identified risk factors for HPV infection were different for different anatomical sites. For anal infection, in addition to the presence of cervical HPV, we identified three very strong risk factors for the prevalence of anal HPV infection (Table 2): ten or more sexual partners in a lifetime (*p* = 0.013), non-coital sexual contact with the anal area (*p* < 0.001), and active practice of anal penetration (*p* < 0.001). For oral infection, the presence of genital warts in the male partner (*p* = 0.029) and history of genital warts in the female partner (*p* = 0.010) were significant risk factors. A study simultaneously evaluating risk factors for anal and oral HPV infections in women with CIN2+ has not yet been published.

Generally, the strongest risk factor for the presence of anal HPV infection is the presence of genital HPV infection. The close relationship between cervical and anal HPV infection was first revealed in a large study in Hawaii on 1378 women. Patients with cervical HPV infection had >3-fold increased risk of concurrent anal infection [6]. Valari found that the presence of cervical HPV was the only statistically significant risk factor for anal HPV with 3.3-fold higher risk [11]. Our HPV-positive CIN2+ patients similarly had a 3.5-fold higher OR (odds ratio) of anal HPV (Table 2). Some authors suggest that women’s risk of anal HPV infection is as common as their risk of cervical HPV infection [7]. These conclusions were also confirmed by studies on high-risk female sex workers [12,13]. A systematic review of 36 studies showed a five-times higher anal HR HPV prevalence among 2693 HIV-negative women with cervical HR HPV infection [14]. A sub-analysis of our data revealed a 4.7-times higher risk of anal HR HPV in women with cervical HR HPV infection (Table 4).

This suggests that the presence of anal HR HPV is an even stronger risk factor for cervical HR HPV. This assumption was confirmed by a recent study that showed that even women successfully treated for CIN may have a higher risk of recurrent cervical HPV infection and cervical lesions due to persistent anal HPV infection [15]. The contemporaneous cervical HPV infection is the most important for increasing the odds of anal infection regardless of whether the cervical infection was detected at a previous visit [8]. However, not only cervical infection but the history of any HPV-related genital lesion elevates the risk of the odds of anal infection. While the prevalence of anal HRHPV among immunocompetent women with HPV-related pathology of the lower genital tract (cervix, vagina, and vulva) varies from 23–86%; among women without known HPV-related pathology, it ranges from only 5–22% [16].

In our cohort, we observed anal HPV in 43.3% (HR HPV in 36.3%) and concurrent cervical-anal HPV infection in 39.3% among women with CIN2+ compared to only 8.0% (*p* < 0.001) among women without CIN2+ ([Fig biomedicines-11-03183-ch001]). Moreover, higher severity of the cervical lesion significantly increased the prevalence of concurrent cervical-anal infections (*p*^trend^ < 0.001). Surprisingly, according to some studies, women with cervical HPV disease have high presence of HR HPV in the anal canal, but contrary to expectations, they also have a very low rate of AIN [17]. Nevertheless, a recent systematic review of twenty-five studies showed that patients after the treatment of HPV-related gynecological diseases have than more five-times higher risk of anal cancer [18].

Several studies have reported that risky sexual behavior is a strong risk factor for anal HPV. In our study, in addition to the presence of cervical HPV infection, we found three other significant risk factors for the prevalence of anal HPV infection (Table 2): ten or more lifetime sexual partners (*p* = 0.013), non-coital sexual contact with the anal area (*p* < 0.001), and active practice of anal penetration (*p* < 0.001). Anal intercourse was noted as a risk factor for anal HPV already in 2001 [19]. Some studies have not confirmed anal intercourse as a risk factor for anal HPV, but there is a general consensus that actively practicing anal-penetrative intercourse increases the risk of anal HPV [13,20]. For example, similar to our study, the prevalence of anal HPV in sexually active women was higher among women who reported anal intercourse (*p* < 0.001), and it increased with a higher number of lifetime sexual partners (*p*^trend^ < 0.001), with a higher number of lifetime anal intercourse partners (*p*^trend^ < 0.001), and with smoking status (*p*^trend^ < 0.001) [8]. Similarly, Goodman revealed a twice-higher odds ratio for anal HPV among women with more than six sexual partners and a three-times higher odds ratio for women with current anal intercourse [7]. Moreover, Goodman found, as we did, that hormone use was a risk factor, too [7]. A high number of sexual partners increased the risk of anal HPV by 2.5 times in a recent international pooled analysis as well, in contrast to practicing anal intercourse [20]. We discovered a history of more than two pregnancies (*p* = 0.045) and more than one delivery (*p* = 0.013) as significant protective factors for anal HPV. It is possible that women with new boyfriends have more frequent anal or oral sex to please their partners but did not report this in the questionnaire. In agreement with current results, sexual non-coital contact with the anus (*p* = 0.006), penetrative anal sex (*p* = 0.002), more than five lifetime sexual partners (*p* = 0.041), and tobacco use (*p* = 0.048) were statistically significant risk factors for multiple anal HPV infections in our former research [21].

While published data show that anal and cervical HPV infections are highly interdependent, data on the association of genitoanal and oral HPV infections are not entirely conclusive [13,22,23,24,25]. Most of the studies following the concurrent cervical or vulvovaginal and oral HPV infections conclude that the relationship with oral genotype–concordance in women with cervical HPV infection is more prevalent than could be expected by chance. Steinau found a five-times higher oral HPV prevalence in 1812 immunocompetent U.S. women with cervical HPV infection [22]. A recent systematic review evaluating 114 papers reported on average a 16% rate of concurrent dual-site oral-cervical HPV infections but with wide differences in particular studies that ranged from 0% to 95% depending on the study [23]. Most rates of concurrent oral and cervical HPV infections were ≤10%. The authors concluded that a definitive conclusion about the association between cervical and oral HPV infection cannot yet be determined. The results of our study were also in line with this. The CIN2+ patients had higher risk of oral HPV in the case of the presence of cervical (OR 1.620, *p* = 0.570) or anal (OR 2.704, *p* = 0.180) HPV infection, but the difference was statistically not significant due to the overall small number of oral HPV-positive cases (Table 3). Nasiotziki, for example, came to the same conclusion on a similar sample of subjects [24]. Kedarisetty detected vaginal HPV infection in 45.2% and oral in 4.1% of 3463 women in the USA. However, more than three-quarters (75.9%) of orally positive women were simultaneously infected in the vagina [25]. These findings correspond with our results. All orally HR-HPV-infected women except one were infected in the cervix or anus, and one-third of them were infected in all three anatomical sites ([Fig biomedicines-11-03183-ch001]).

The presence of genital warts in the male partner (*p* = 0.029) and a history of genital warts in the woman (*p* = 0.010) were found statistically significant for the prevalence of oral HPV infection in our study (Table 3). Surprisingly, active oral sex was not a risk factor for oral infection. Bruno indicated oral sex as a risk factor for oral HPV infection (*p* < 0.01) in his paper, however, with less statistical significance than a history of genital warts (*p* < 0.001) [26]. This means that simply practicing oral sex is not enough, but other risk factors for acquiring of oral HPV must be present. The reason genital warts in one sexual partner is a risk factor for oral HPV seems to be that oral sex is practiced by almost all sexually active couples these days, e.g., in France, the number of women reporting active oral sex practice increased from 51% in 1970 to 91% in 2006. Similarly, an increase in male oral sex practice over the same period was noted from 55% to 94% [27]. On the other side, risky sexual behavior, tobacco use, alcohol use, drug use, and the presence of other sexually transmitted diseases except HIV were not proven to be risk factors for the presence of cervical, anal, or oral HPV infections in the study among 315 female sex workers from Nigeria [13]. However, most earlier studies reported different results [10,28,29].

Age plays a greater role in the risk of anal HPV infection prevalence than in oral HPV prevalence and probably than in cervical HPV prevalence. Hernandez found that age < 30 increased the risk for anal HPV [9], and Goodman found that age > 45 decreased the likelihood of anal HPV [7]. We found in our study that women aged above 50 years were significantly associated with lower prevalence (*p* = 0.048) of anal HPV infection. A large analysis of 36 longitudinal studies revealed a decreasing prevalence of anal HPV infection every 10 years of about 0.85 times [20].

The overall oral HPV prevalence was quite low in our study (2.7% in the HR group, 1.4%, in the LR group, and 2.3% in the whole cohort). However, HR genotypes were detected in all HPV-positive oral samples of patients with CIN2+. On the other hand, this is not a significant difference from the prevalence of oral infection of 2.9% in women (two-thirds lower than in men of the same age) calculated in the meta-analysis [29] and from the 2.5% oral HPV prevalence among at-risk women referred for colposcopy in other paper [24]. According to a systematic review of 28 papers, the overall oral HPV prevalence rates in healthy European populations is between 1.2% and 11.6% [10]. HPV infection in the oral cavity can be quite difficult to detect and easy to miss. Virus clearance is supported by continuous rinses of saliva in the mouth, and this makes oral HPV more transient than in other anatomical sies [23].

Another reason may be the generally lower prevalence of oral HPV in White women. A cross-sectional study from the USA reported the lowest prevalence of oral HPV in White women compared with Mexican, Hispanic, and Black women and with men of any race [30]. Another reason for the low oral prevalence may be also due to the use of oral rinses to detect HPV in our study. Oral HPV prevalence was lower (mean 4.7%) in studies using an oral rinse than in studies using a smear (mean 10.3%) [10].

There are also some limitations that should be noted. This study was performed on a homogeneous population of Czech women, and the results do not indicate global HPV prevalence and the associated risk factors. Our study assumed a cross-sectional setting, and it could not be determined how many of the HPV infections were transient. The Linear Array HPV Genotyping Test used in our study was designed for genotyping of cervical HPV infection, and therefore, true oral HPV prevalence may be slightly underestimated [30,31]. Another potential limitation is the lack of data regarding oral, anal, and genital HPV infections in male sexual partners, especially when the presence of genital warts in sexual partners was a significant risk factor for oral HPV infections. The prevalence of genital HPV infection among the male sexual partners of HPV-infected women seems to be independent of the length of their relationship, and it appears to be very high [32].

Furthermore, we did not ask about some kinds of another possible routes of HPV transmission. For example, the cervix and anus can also be infected during masturbation due to autoinoculation [8]. Moreover, open-mouth kissing with oral-to-oral contact is considered a significant risk factor for oral HPV infection [28]. Additionally, self-reported sexual behaviors may not be accurate. We did not examine other infectious diseases, although an association with coinfection by other sexually transferred diseases is independently associated with HPV infection in multivariable models [31]. Last, we did not ask about anal fissures, which could significantly increase the odds of anal HPV infection among women who do not report anal sex [17].

On the other hand, this is the first study that examines not only the prevalence but also the risk factors for anal and oral HPV infections in high-risk women with CIN2+. An important strength of our study is that all women were prospectively recruited. First, the women filled out a self-administered questionnaire, and then, samples were taken for cervical, anal, and oral HPV genotyping at the same time. Nevertheless, other prospective studies are needed to clarify the risk factors for HPV infections at all HPV-related anatomical sites to define preventive programs, adequate screening, and vaccination strategies.

## 5. Conclusions

In conclusion, our data suggest that the presence of HPV infection in one anatomical site poses the greatest risk for HPV infection in another anatomical site. The presence of cervical HR HPV infection seems to be an even stronger risk factor for anal HR HPV infection and possibly also for oral HR HPV infection. We hypothesize that the cervix is the main reservoir of infection, but the risk factors for the anal and oral HPV infections are dissimilar. Transmission from the anogenital region to the oral region is likely to be more difficult due to anatomical distance and more complex routes of transmission. However, a separate oral infection without concurrent genitoanal infection is very rare. Sexual behavior, including autoeroticism and non-coital sex, is of great importance. Moreover, our results may reflect differences in the natural course of infections in different anatomical sites and thus contribute to the development of preventive strategies because no formal general recommendations exist regarding HPV-related cancer screening except for cervical cancer so far.

## Data Availability

All data generated or analyzed during this study are included in this published article.

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
