# Peer review of "Risk Factors for the Anal and Oral Human Papillomavirus (HPV) Infections among Women with Severe Cervical Lesions: A Prospective Case—Control Study"

_biomedicines, 2023, doi:10.3390/biomedicines11123183_

Round 1

Reviewer 1 Report

Comments and Suggestions for Authors

The authors aimed to define risk factors  for the presence of anal and oral HPV infections in high-risk patients with biopsy-proven  severe cervical lesions.

 It is better not to abbreviate HPV in the title..

The authors should mention how the sample size was calculated.

 What is the relation of the prevalence of concurrent cervical-anal infection to the grade of cervical lesion?

 Please refer to line 248. Contrary to what authors say, an earlier study reported that alcohol use increases the risk (Reference- Smith EM, Rubenstein LM, Haugen TH, Hamsikova E, Turek LP. Tobacco and alcohol use increases the risk of both HPV-associated and HPV-independent head and neck cancers. Cancer Causes Control. 2010 Sep;21(9):1369-78.).

Tobacco/alcohol cause genetic damage at multiple foci in the head and neck region.

 It is better to convey both the facts i.e., either alcohol is a risk factor and also, it is not.

 Did the women declare homosexual identity?

 Can sociocultural factors such as increasing immigration rates, rising sexual promiscuity, and immigration status influence the results?

 What was the level of sexual education in the subjects? Could such influence the results?

 Is it advisable to observe HPV-DNA?

 Did the authors look into recurrence? Can the tumor stage influence the rate of recurrence?

 Low-risk HPV types, e.g. HPV6 and 11 are associated with upper airway papillomas. Did the authors look into this?

 Degradation of archival tumor specimens may contribute to the underestimation of HPV prevalence in tumors. The authors may discuss these facts.

Comments on the Quality of English Language

Minor corrections are needed.

Author Response

Reviewer 1

Comments and Suggestions for Authors

The authors aimed to define risk factors  for the presence of anal and oral HPV infections in high-risk patients with biopsy-proven  severe cervical lesions.

Answer: We thank to reviewer for targeted comments.   Borek Sehnal et al.

 It is better not to abbreviate HPV in the title..

Answer: The title was changed: Risk Factors for the Anal and Oral Human Papillomavirus (HPV) Infections among Women with Severe Cervical Lesions; a prospective case-control study.

The authors should mention how the sample size was calculated.

Answer: The statistician was not consulted in advance to indicate the minimum number of samples necessary for statistical processing. Moreover, the percentages of common HPV infections were calculated from samples sufficient for HPV testing, therefore the number of subjects planned by the statistician would probably not correspond to the final results.

What is the relation of the prevalence of concurrent cervical-anal infection to the grade of cervical lesion?

Answer:  The prevalence of concurrent cervical-anal infection  significantly increased with the higher grade of cervical lesion (CIN 1 vs. CIN 2 vs. CIN 3 ptrend<0.001)

We add to the manuscript: “Higher severity of cervical lesion significantly increased a prevalence of concurrent cervical-anal infections (ptrend<0.001, data not shown) “

Please refer to line 248. Contrary to what authors say, an earlier study reported that alcohol use increases the risk  

(Reference- Smith EM, Rubenstein LM, Haugen TH, Hamsikova E, Turek LP. Tobacco and alcohol use increases the risk of both HPV-associated and HPV-independent head and neck cancers. Cancer Causes Control. 2010 Sep;21(9):1369-78.).

Tobacco/alcohol cause genetic damage at multiple foci in the head and neck region

 It is better to convey both the facts i.e., either alcohol is a risk factor and also, it is not.

Answer: The reviewer is absolutely right. It is generally accepted, that consumption of alcohol is an important risk factor, which might facilitate oral viral infection by interfering with the integrity of the oral mucosa or with homeostatic balance in the buccal cavity.  Most work suggests that alcohol and tobacco use increase the risk of oral HPV infection. But that's a recent quote from Q1 journal where they didn't confirm it. This probably means, if other risk factors prevail, even the use of tobacco, alcohol and drugs may not appear as major risk factors. But to be more objective, we added to the manuscript: “However, most earlier studies reported different results [Smith, Cancer Causes Control. 2010; Shigeishi, J Clin Med Res. 2016; Simonidesová Epidemiol Mikrobiol Imunol. 2018]. “

 Did the women declare homosexual identity?

Answer: No, the women did not declare homosexual identity. Questions about sexual orientation and sexual behavior were in a self-administered questionnaire. Any woman did not declare a homosexual identity

Can sociocultural factors such as increasing immigration rates, rising sexual promiscuity, and immigration status influence the results

Answer: We think not. The Czech Republic is a country with very low immigration. The questionnaire was in Czech and it is a very difficult language. Only Czech women who were able to fill in the self-administered questionnaire participated in the study. We also do not think that sexual promiscuity rise in the Czech Republic.

 What was the level of sexual education in the subjects? Could such influence the results?

Answer:  The level of sex education in the Czech Republic is very high. For example, from the neighboring country of Poland, only a few people are actively practicing Christians. Sex education is usually talked about shamelessly. Therefore, we think that the level of sex education did not affect the results.

 Is it advisable to observe HPV-DNA?

Answer: We believe so. HPV testing catches more infections than HPV mRNA testing, so we think that HPV DNA testing has significant impact in this type of study.

Did the authors look into recurrence? Can the tumor stage influence the rate of recurrence?

Answer: It is good question. We did not follow HPV recurrence. This can be topic for another study.

 Low-risk HPV types, e.g. HPV6 and 11 are associated with upper airway papillomas. Did the authors look into this?

Answer:  Unfortunately, no. The aims of this study were different. The patients were not even specifically examined for the presence of upper airway papillomas. For these reasons, we cannot discuss this topic, even though this topic is very interesting.

 Degradation of archival tumor specimens may contribute to the underestimation of HPV prevalence in tumors. The authors may discuss these facts.

Comments on the Quality of English Language: Minor corrections are needed.

Submission Date 15 October 2023

Date of this review  20 Oct 2023 16:31:01

Reviewer 2 Report

Comments and Suggestions for Authors

Author Response

Author's Notes to Reviewer are attached.

Round 2

Reviewer 2 Report

Comments and Suggestions for Authors

Thank you for your revision but I would like to bring to your attention some points not considered.

-           ABSTRACT: How said in the abstract it not supported by results

“Our data suggest that the presence of HPV infection, especially high-risk genotypes, in one anatomical site poses the greatest risk for HPV infection in another anatomical site”

-           In DISCUSSION the authors said "Subanalysis of our data revealed a 4.7 times higher risk of anal HR HPV in women with cervical HR HPV infection (data not shown)."

The authors didn't show the results for high/low risk human papillomavirus genotypes and this is a crucial point.

To solving the problem of limited by the maximum number of tables you can unite table 2 and table 3. They have the same first column (risk factors)

Please insert table to describe LR/HR HPV anal infections and LR/HR HPV oral infections and show the results for high/low risk human papillomavirus genotypes. Please reported the co-analysis of anal HR HPV in women with cervical HR HPV infection.

 Thanks for the information reported in the cover letter but there are not presented in paragraph "2.4. Statistics. Please insert the information reported in the cover letter.

Author Response

Due to the comments of reviewer number 2, we have made the following changes in the manuscript:

We inserted this text to a paragraph 2.4. Statistics

“The standard robust summary statistics were applied to describe primary data, absolute and relative frequencies for categorical variables and median supplied with the 5th-95th percentile range for continuous variables. The statistical significance of differences between group of the subjects with anal or oral HPV infections and group of the subjects without anal or oral HPV infections was tested in categorical variables using Fisher exact test; an exact Monte Carlo method with 100 000 samples was applied to estimate the significance of differences in variables with more than two categories. Mann-Whitney U test was applied to test the differences in continuous variables. Age-adjusted logistic regression analysis was used for the assessment of the association between various risk factors and defined end-points related to different types of HPV infection. The results are represented as estimates of odd ratios (OR, along with 95% confidence interval) with corresponding statistical significance (Wald’s test).  All analyses were performed using SPSS 24.0.1. (IBM Corporation, 2016). We expanded paragraph "2.4. Statistics.”

We added Table 4: Risks for concurrent HPV infections among women with CIN2+

Round 3

Reviewer 2 Report

Comments and Suggestions for Authors

Thank you for your work.

Author Response

I thank the reviewer for targeted comments, which certainly increased the quality of the manuscript.

Borek Sehnal